# MPC Polymer Promotes Recovery from Dry Eye via Stabilization of the Ocular Surface

**DOI:** 10.3390/pharmaceutics13020168

**Published:** 2021-01-27

**Authors:** Noriaki Nagai, Shunsuke Sakurai, Ryotaro Seiriki, Misa Minami, Mizuki Yamaguchi, Saori Deguchi, Eiji Harata

**Affiliations:** 1Faculty of Pharmacy, Kindai University, 3-4-1 Kowakae, Higashi-Osaka, Osaka 577-8502, Japan; 1611610157u@kindai.ac.jp (R.S.); 2033420004w@kindai.ac.jp (M.M.); 2033420005s@kindai.ac.jp (M.Y.); 2045110002h@kindai.ac.jp (S.D.); 2Life Science Products Division, NOF Corporation, Yebisu Garden Place Tower, 20-3 Ebisu 4-chome, Shibuya-ku, Tokyo 150-6019, Japan; shunsuke_sakurai@nof.co.jp (S.S.); eiji_harata@nof.co.jp (E.H.)

**Keywords:** MPC polymer, dry eye, ocular surface, lacrimal fluid, mucin

## Abstract

The polymer that includes 2-methacryloyloxy ethyl phosphorylcholine (MPC) is well-known as an effectively hydrating multifunction agent. In this study, we prepared an MPC polymer (MPCP) using radical polymerization with co-monomers—MPC/Stearyl Methacrylate/*N*,*N*-dimethylacrylamide—and evaluated the MPCP’s usefulness for dry eye treatment using a rabbit model treated with *N*-acetylcysteine. The MPCP particle size was 50–250 nm, and the form was similar to that of micelles. The MPCP viscosity (approximately 0.95 mPa·s) was 1.17-fold that of purified water, and a decrease in the transepithelial electrical resistance value (corneal damage) was not observed in the immortalized human corneal epithelial cell line HCE-T cell (HCE-T cell layer). The MPCP enhanced the water maintenance on the cornea, and the instillation of MPCP increased the lacrimal fluid volume and prolonged the tear film breakup time without an increase in total mucin contents in the lacrimal fluid of the normal rabbits. The therapeutic potential of the MPCP for dry eye was evaluated using an *N*-acetylcysteine-treated rabbit model, and, in our investigation, we found that MPCP enhanced the volume of lacrimal fluid and promoted an improvement in the tear film breakup levels. These findings regarding the creation and characteristics of a novel MPCP will provide relevant information for designing further studies to develop a treatment for dry eyes.

## 1. Introduction

Dry eye disease is a complex and multifactorial ocular disease, and tear film instability, visual disturbance, and discomfort are commonly observed, with potential damage to the ocular surface [1]. It was reported that dry eye disease has multiple causes, such as air pollution, medication, androgen deficiency, contact lens usage, and excessive computer use [2]. Inflammation and enhanced osmolarity on the ocular surface are also important factors in the onset of dry eye disease [3]. It has also been reported that oxidant stress and aging are other factors that contribute to dry eye disease [4]. The prevalence of dry eye disease is estimated at approximately 5–50% of the global adult population [5], and it is expected that the economic burden from the disease will increase as the population ages [6].

Current treatments for dry eye disease aim to prevent the objective signs and clinical symptoms of the disease and recover the quality of life of the patients. The main approved treatments are as follows: Mucosta^®^ (rebamipide ophthalmic suspension 2%, Otsuka Pharmaceutical, Tokyo, Japan) and Diquas^®^ (diquafosol ophthalmic solution 3%, Santen Pharmaceutical, Osaka, Japan) in Asia; Xiidra^®^ (lifitegrast ophthalmic solution 5.0%, Shire, Lexington, KY, USA) and Restasis^®^ (Cyclosporine ophthalmic emulsion 0.05%, Allergan, Irvine, CA, USA) in North America; and Ikervis^®^ (CsA cationic emulsion, Santen Pharmaceutical, Osaka, Japan) in Europe. In addition, Cequa^TM^ (ananomicellar formulation of CsA 0.09%, Sun Pharmaceuticals, Mumbai, India) has been used in the US since 2018. Cequa^TM^ enhances lacrimal fluid production in patients with dry eye disease. Overall, these medicines are used to stabilize the tear film and/or prevent the inflammation of the ocular surface [7].

The polymer including 2-methacryloyloxy ethyl phosphorylcholine (MPC) is well-known for being not only an antiadhesive and antithrombogenetic agent, but also a significant hydrating multifunction agent [8,9,10]. These characteristics are due to the polymer’s phosphorylcholine group and the water structure surrounding the polar group. To date, soft contact lenses containing this MPC have been approved by the Food and Drug Administration (FDA) and Pharmaceuticals and Medical Devices Agency (PMDA). Hall et al. have reported that the dehydration of this soft contact lens is significantly less than that of other conventional lenses [11]. Eye drops, soft contact lens care products, oral care products, and cosmetics containing the MPC-BMA polymer have also been approved by the PMDA. Ayaki et al. reported that, after treatment with eye drops containing MPC polymer (MPC-BMA), cell viability rates were maintained at over 80%. Moreover, the conformation of proteins did not change, even when they were adsorbed on the surface or came into contact with the surface, and were similar to those of clinically approved artificial tear products [12]. In this way, the polymer including MPC, which is significantly hydrating and safe, could be used to improve the ocular surface and treat dry eye disease with the potential to be added to dry-eye drugs.

In this study, we prepared a novel MPC polymer using radical polymerization with co-monomers, MPC/stearyl methacrylate/*N*,*N*-dimethylacrylamide (MPCP), and evaluated the polymer’s usefulness for dry eye treatment using a rabbit model treated with *N*-acetylcysteine.

## 2. Materials and Methods

### 2.1. Animals

Male adult rabbits (weight 2.58 ± 0.75 kg) were obtained from Shimizu Laboratory Supplies Co., Ltd. (Kyoto, Japan), and the protocol was approved by the Kindai University (KAPS-31-002, 1 April 2019). The experiments using the rabbits were performed according to the Association for Research in Vision and Ophthalmology (ARVO) and Kindai University guidelines. Thirty microliters of 0.1% MPCP were instilled in single applications at 14:00 h. For repeat applications, 0.1% MPCP (30 µL) was instilled once a day (14:00 h) for 5 days, and the measurement of the levels of lacrimal fluid, mucin, tear film breakup time (TBUT), ocular surface, and tear film breakup began at 16:00 h (single application) and 18:00 h (repetitive application). The dry eye model using rabbits was conducted via the instillation (30 µL) of 10% *N*-acetylcysteine six times per day (at 9:00, 11:00, 13:00, 15:00, 17:00, and 19:00 h). This protocol was performed following our previous report [13].

### 2.2. Chemicals

Dulbecco’s modified Eagle’s medium/Ham’s F12, penicillin, streptomycin, and fetal bovine serum were provided from GIBCO (Tokyo, Japan). Cell Count Reagent SF was purchased from Nacalai Tesque, Inc. (Kyoto, Japan), and the Tear mucin assay ELISA kit was obtained from Cosmo Bio Co., Ltd. (Tokyo, Japan). Transwell-Clear^TM^ (polyester filters, surface area 1.0 cm^2^ and 0.4 μm pore size) and rat tail collagen type 1 were purchased from Costar (Cambridge, MA, USA) and Sigma (Tokyo, Japan), respectively. All other chemicals used were of the highest purity commercially available.

### 2.3. Preparation of MPCP

MPCP was obtained via radical polymerization with co-monomers, MPC/stearyl methacrylate/*N*,*N*-dimethylacrylamide, with a composition ratio of 50:5:45, and purified using the dialysis method. After polymerization, we measured the residual monomers and calculated that the conversion rate of this polymer in each monomer was >99%. We thus confirmed that the target polymer was obtained. In this study, we used MPCP diluted to a 1% aqueous solution with purified water. To confirm the MPCP concentration of the aqueous solution, we studied the residue upon drying. The structural formula of the MPCP is illustrated in Figure 1.

### 2.4. Measurement of Characteristics in MPCP

A NANOSIGHT LM10 (Quantum Design Japan, Tokyo, Japan) was used to measure the size and number of MPCP nanoparticles—the measurement was performed for 60 s at 405 nm. Atomic force microscopy (AFM) images were obtained via an A SPM-9700 (Shimadzu Corp., Kyoto, Japan), and AFM images were created by combining the phase and height images. The viscosity of MPCP was measured with an SV-1A at 10–40 °C (A&D Company, Limited, Tokyo, Japan) [14]. To confirm the wettability, defined as the tendency of one fluid to spread on or adhere to a solid surface, we performed contact angle measurements in 0.1% MPCP and 0.1% MPC-BMA aqueous solutions. In total, 1 µL of each of the 0.1% MPCP and 0.1% MPC-BMA aqueous solutions (or purified water as a control) was dropped onto a slide glass, and the contact angle was measured using a contact angle meter (DropMaster500, Kyowa Interface Science Co., Ltd., Saitama, Japan).

### 2.5. Cell Culture and Treatment

The immortalized human corneal epithelial cell line (HCE-T cell) used in this study were developed by Araki-Sasaki et al. [15]. The HCE-T cells were cultured in Dulbecco’s modified Eagle’s medium/Ham’s F12 with heat-inactivated fetal bovine serum (5%), penicillin (1000 IU/mL), and streptomycin (0.1 mg/mL).

### 2.6. Measurement of Cell Adhesion

The MPCP treatment was carried out by seeding HCE-T cells (1 × 10^4^ cells) in a culture medium containing 0.1% MPCP and incubating the medium for 12 h. Then, a Cell Count Reagent SF was added, and the absorbance (Abs) at 450 nm was measured according to the manufacturer’s protocol. The Cell Count Reagent SF is based on the conversion of the reagent to formazan salts according to mitochondrial activity and, as a metabolic assay, is related to the cell number and its metabolic state and efficiency. Therefore, the cell number was also measured via counting under a microscope (Olympus Corporation, Tokyo, Japan), and the cell adhesion was evaluated using a combination of data on the cell number and Cell Count Reagent SF. The cell adhesion (%) was represented as the Abs ratio of the MPCP treatment and non-treatment [16].

### 2.7. Measurement of Cell Proliferation

The cell cultures were treated with MPCP 1 d after seeding (1 × 10^4^ cells) by changing to a culture medium with 0.1% MPCP and incubating the medium for 24 h. Then, the Cell Count Reagent SF was added, and the Abs at 450 nm was measured according to the manufacturer’s protocol. Cell proliferation (%) was recorded as the Abs ratio of the MPCP treatment and non-treatment [16]. The cell proliferation was also evaluated by a combination of the data on the cell number and Cell Count Reagent SF in this study.

### 2.8. Preparation of HCE-T Cell Layer Model

The cell layer models (multilayer) consisting of one cell (only HCE-T cells) were cultured by following our previous reports [17,18]. The HCE-T cells were seeded onto Transwell-Clear^TM^ (90,000 cells/cm^2^) coated with rat tail collagen type 1 (71.5 µg/cm^2^) and grown for seven days until the cells reached confluency. The HCE-T cells were then exposed to an air–liquid interface for two weeks with the culture medium containing the vehicle or 0.1% MPCP replaced every other day. In this process, 50 µL of the medium solution with or without MPCP was dropped onto the donor side twice a day (9:00 and 19:00 h) so that the donor would not dry completely. In this study, chopstick electrodes connected to an epithelial Volt–Ohm meter Millicell-ERS (Millipore Co., Bedford, MA, USA) were used to measure the transepithelial electrical resistance (TER) and followed the differentiation stages during cultivation.

### 2.9. Cell Toxicity of MPCP

Differentiated cells with TER values greater than 350 Ω·cm^2^ were used for the cell toxicity analysis. The HCE-T cell layer model was treated with the vehicle and 0.1% MPCP, and the changes in TER values were measured for 60 min [17].

### 2.10. Measurement of Water Retention in the Cornea

The rabbits were euthanized by injecting a lethal dose of pentobarbital, and the corneas were carefully separated from other ocular tissues. The individual corneas were treated with the vehicle, 0.1% MPC-BMA, and 0.1% MPCP for 1 min. After that, the cornea was placed on a plastic cell, and the changes in the weight and number of MPCP particles were measured at 22 °C. In this study, the changes in weight were expressed as water retention in the cornea. Moreover, the samples (vehicle, MPC-BMA, and MPCP) on the cornea were collected by pipette, and the number of MPCP particles in the samples was measured with NANOSIGHT LM10, as described above.

### 2.11. Monitoring the Ocular Surface of Rabbits Instilled with MPCP

The ocular surface was monitored by using a dry eye monitor DR-1 (KOWA Co., LTD., Aichi, Japan), and the TBUT and changes in the ocular surface were monitored following our previous study [13]. Each rabbit treated with a fluorescein strip was allowed to blink several times to distribute the fluorescein. The time from the opening of the eyes to the appearance of the first dry spot in the central cornea was analyzed. The changes in tear film after blinking were monitored by the dry eye monitor DR-1. The TBUT and the tear film breakup level changes in the ocular surface were evaluated 2 h (16:00 h) and 4 h (18:00 h) after the application of the MPCP, respectively. The tear film breakup levels (area) were measured 2 s after the last blink using the Image J software (ver. 1.51, NIH, USA), and the measurement was performed three times; the mean was used as the value.

### 2.12. Lacrimal Fluid and Mucin Levels in Rabbits Instilled with MPCP

The volume of lacrimal fluid in rabbits instilled with MPCP was measured using Schirmer tear test strips (AYUMI Pharmaceutical Corporation, Tokyo, Japan), and the mucin was measured using a Tear mucin assay ELISA kit according to the manufacturer’s instructions. Briefly, lacrimal fluid was collected with Schirmer tear test strips, and the Schirmer tear test strips were added into elution buffer of a tear mucin assay ELISA kit to extract the mucin. After that, the extracted mucin was measured by the Tear mucin assay ELISA kit, and a fluorescence microplate reader (Absorption/Emission = 336/383 nm) [13]. The mucin levels in the total lacrimal fluid volume of the eye are expressed as total mucin content (µg). The mucin concentration in lacrimal fluid (mg/mL) is estimated from the mucin level/lacrimal fluid volume.

### 2.13. Statistical Analysis

Differences between the mean values were analyzed with an ANOVA, followed by Student’s *t*-test, and Dunnett’s multiple comparisons test was used for the statistical analysis, where *p* < 0.05 was considered significant. The data are expressed as the mean ± standard error (S.E.).

## 3. Results

### 3.1. Design of the MPCP

Figure 1 shows the structural formula of the MPCP. We designed the MPCP structure based on three points: (1) the MPC, indicating the zwitterionic group, shows the hydrophilic part and gives hydrophilicity to MPCP; (2) stearyl methacrylate including the long-chain alkyl group shows its hydrophobic part and forms a hydrous polymer nano-sphere in an aqueous solution; (3) coupled with the acryl group as a highly reactive functional group, *N*,*N*-dimethylacrylamide accelerates the polymerization between MPC and stearyl methacrylate and strengthens the structure of polymer nano-sphere in an aqueous solution. Figure 2A,B shows the particle distribution and an AFM image of the MPCP. This polymer includes monomer forms similar to micelles (polymer nanosphere), with a hydrophilic outer-most layer and a hydrophobic inner-most layer in an aqueous solution, featuring a particle size of from 50 to 250 nm. In addition, the particle number was 349 ± 0.336 (×10^8^ particles/mL). Figure 2C,D shows the viscosity of MPCP at 10–40 °C. The viscosity of MPCP was 1.17- and 0.92-fold that of the purified water and MPC-BMA reported previously (preexisting MPC polymer) [12], respectively, and the viscosity levels of MPCP were similar under 10–40 °C conditions. We also measured the hydrophilicity of MPCP and MPC-BMA. The contact angle using MPCP was 25.5 ± 1.3° (*n* = 3), the contact angle using MPC-BMA was 26.2 ± 2.0° (*n* = 3), and the control was 29.2 ± 0.4° (*n* = 3). These results indicate that the MPCP was slightly hydrophilic and had a higher wettability than the MPC-BMA and the control.

### 3.2. Changes in Cell Conditions in the Immortalized Human Corneal Epithelial Cell Line (HCE-T Cell) Treated with MPCP

Figure 3A,B shows the effects of MPCP treatment on cell adhesion and growth in the HCE-T cells. No significant differences were observed between the cell adhesion and growth of HCE-T cells treated with and without MPCP. Figure 3C shows the effect of MPCP treatment on the TER during the HCE-T cell layer model preparation. The TER values of HCE-T cells increased to over 380 Ω·cm^2^ when we exposed them to an air–liquid interface for two weeks. The TER in the MPCP-treated HCE-T cells was similar to non-treated values before exposure to the air–liquid interface; however, during exposure to the air–liquid interface, the TER values in the MPCP-treated HCE-T cells were significantly higher than those found in the HCE-T cells without MPCP treatment. Figure 3D shows the cell toxicity caused by MPCP treatment. The cell toxicity caused by the MPCP was mild, since the decrease in TER in the MPCP-treated HCE-T cells was similar to that in the vehicle-treated HCE-T cells.

### 3.3. Effect of MPCP on the Ocular Surface Stability in the Normal Model

As shown in Figure 3, the treatment with MPCP enhanced cell stability in the HCE-T cell layer model. We investigated whether the ocular surface stability in a normal rabbit was increased via the instillation of MPCP. The lacrimal fluid volume and TBUT were significantly increased via treatment with MPCP (Figure 4A,D); however, the mucin concentration in the lacrimal fluid decreased in the MPCP-instilled rabbits; for the total mucin levels in the lacrimal fluid, no difference between the vehicle- and MPCP-treated groups was found (Figure 4B,C). On the other hand, the lacrimal fluid volume (18.4 ± 1.9 µL, *n* = 5) and TBUT (24.1 ± 2.0 s, *n* = 5) in the rabbits instilled with MPC-BMA tended to increase, although both the lacrimal fluid volume and TBUT were significantly lower than those in the MPCP. Figure 5 shows the effects of MPCP on moisture retention in the excised rabbit cornea. The water content in the cornea instilled with the vehicle decreased with time and completely evaporated after 90 min. The MPCP instillation prolonged the time needed for water evaporation from the cornea and the number of MPCP nanoparticles to decrease on the cornea. Preexisting MPC-BMA prolonged the time needed for water maintenance in comparison with the vehicle, but this ability was significantly lower than that in MPCP.

### 3.4. Therapeutic Potential of the MPCP for Dry Eye Disease

Next, we investigated the usefulness of MPCP as a therapy for dry eye using the *N*-acetylcysteine-treated rabbit model (Figure 6). The levels of lacrimal fluid and mucin decreased under treatment with *N*-acetylcysteine, with levels 0.8- and 0.67-fold greater than those of normal rabbits, respectively (Figure 6A,B). Both the lacrimal fluid volume and mucin levels were approximately 2.5-fold greater in comparison with the control group (non-instillation group) (Figure 6A,B). Figure 6C,D show the changes in the levels of tear film breakup in the *N*-acetylcysteine-treated dry eye model rabbits instilled with (or without) MPCP. Strong tear film breakup levels were observed and still persisted five days later. Repeat treatment with MPCP attenuated the tear film breakup levels five days post-*N*-acetylcysteine treatment, and the therapeutic effect of MPCP was significantly higher than that in MPC-BMA (tear film breakup levels at 5 days, 4.26 ± 1.09 mm^2^, *n* = 5).

## 4. Discussion

In this study, we designed a novel MPCP that includes the zwitterionic group stearyl methacrylat, and the acryl group, and found that the MPCP has a high affinity for water and moisturizes the ocular surface in comparison with the MPC-BMA reported previously (preexisting MPC polymer) [12]. We also showed that the instillation of MPCP appears to provide a useful therapy for dry eye (Figure 7).

First, we designed MPCP and evaluated its characteristics. MPC/stearyl methacrylate/*N*,*N*-dimethylacrylamide was used to prepare the MPCP in this study, and MPCP was purified using the dialysis method. The over-30-nm nanoparticles reported previously were not detected in the MPC-BMA [12]. In contrast with the results for MPC-BMA, this polymer (MPCP) included monomer forms similar to micelles (polymer nanosphere), with particle sizes of 50–250 nm (Figure 2A,B). The viscosity of MPCP was 0.92-fold that of MPC-BMA and was not different under 10–40 °C conditions (Figure 2D). On the other hand, the contact angle value of MPCP tended to decrease compared to that of the MPC-BMA. These results suggest that the MPCP has different physicochemical properties from preexisting MPC-BMP due to the formation of nanoparticles.

Next, we investigated the effects of MPCP on corneal epithelial cells using HCE-T cells. We previously reported that MPC-BMA does not affect the cell adhesion and growth of HCE-T cells, with little toxicity in HCE-T cells [19]. The MPCP did not affect the cell adhesion and growth of the HCE-T cells (Figure 3A,B), and no cell toxicity was observed (Figure 3D); however, we know that the barrier properties of the HCE-T cell layer model are very similar to those in rabbit corneas, and an increase in the barrier properties of the HCE-T cell layer model is expressed as an enhancement of the TER value [15,18]. To prepare the HCE-T cell layer model, the cells were exposed to air in order to increase the TER value, as this step was important to instill the medium solution to the cells twice a day (9:00 and 19:00 h), ensuring that the cell layer model would not dry completely. As a result, we demonstrated that the stabilization of the cell layer model increased via the instillation of a drop of MPCP to the therapeutic target. The MPCP enhanced the TER values of HCE-T cells during exposure to the air–liquid interface (Figure 3C). Taken together, the MPCP may attenuate the desiccation on the donor side, resulting in the enhancement of the TER value in the HCE-T cell layer model.

Further, we investigated the changes in the ocular surface in normal rabbits instilled with MPCP (Figure 4). The instillation of MPCP enhanced the lacrimal fluid volume and prolonged the TBUT (Figure 4A,D). These results show that the instillation of MPCP stabilized the ocular surface conditions; thus, the mucin level relates to the increase in lacrimal fluid and TBUT. The mucins composed of numerous sugar chains linked to an apomucin, which is a core protein, were heavy molecular glycoproteins, and 50–80% of their mass was comprised of carbohydrates. It was reported that mucin leads to the formation of a smooth spherical surface, offering good vision, the provision of a barrier for the ocular surface, lubrication of the ocular surface to facilitate smooth blinking, and maintenance of the lacrimal fluid on the ocular surface [20,21]. Moreover, the mucin layer spreads lacrimal fluid over the surface of the eye by decreasing the surface tension of the water content. In addition, the mucin can remove foreign materials and prevent damage and infections in the eye [22]; therefore, an increase in mucin may enhance the lacrimal fluid volume and TBUT. Based on previous studies, we also measured the mucin levels in the lacrimal fluid of normal rabbits instilled with MPCP. Against all expectations, the total mucin content in the lacrimal fluid of the MPCP-instilled rabbits was similar to that of the vehicle-instilled rabbits, and the concentration of lacrimal fluid of the MPCP-instilled rabbits was lower than that of the vehicle-instilled rabbits (Figure 4B,C). These results suggest that MPCP is not affected by mucin production, since the total mucin levels were similar in rabbits instilled with or without MPCP. On the other hand, the mucin levels may be diluted by the enhanced lacrimal fluid volume via MPCP instillation, resulting in an apparent reduction in the mucin concentration in lacrimal fluid. To understand this result, we investigated whether MPCP enhanced the moisture retention in the excised rabbit cornea, and the results showed that the MPCP instillation prolonged the time needed for water evaporation from the cornea (Figure 5A,B). Although the preexisting MPC-BMA [12] also prolonged the time needed for water maintenance in comparison with the vehicle, this ability was significantly lower than that in MPCP. These results suggest that the preexisting MPC-BMA and MPCP exert similar effects. However, the MPCP has a strong ability to retain water on the cornea in comparison with MPC-BMA, and this high water affinity may enhance the lacrimal fluid volume and TBUT in normal rabbits. Moreover, this water affinity on the cornea may be enhanced by the formulation of hydrous polymer nano-spheres and polymerization of the MPC, since the time needed for water maintenance was prolonged in comparison with the MPC-BMA.

We found that the number of MPCP nanoparticles decreased on the cornea after treatment (Figure 5C). The MPCP may bind to the cornea, resulting in a decrease in forms like micelles in the solution. Further studies are needed to demonstrate the binding of MPCP on the cornea.

It is important to elucidate the therapeutic effect of MPCP in a dry eye model. The *N*-acetylcysteine, a reducing agent and mucolytic [23], acts on corneal and conjunctival epithelial cells and decreases mucin production and retention. Further, the instillation of 20% *N*-acetylcysteine caused a decrease in the mucin layer in the conjunctiva and cornea, desquamation of conjunctival and corneal epithelial cells, elimination of microvilli [24], and decreased thickness of the tear fluid layer [25]. Because of these actions, a rabbit model treated with *N*-acetylcysteine is widely applied as a mucin-reduced model and may be useful for evaluating the therapeutic effect of the MPC polymer for dry eye. Therefore, we treated a rabbit model with *N*-acetylcysteine and measured the effect of MPCP on the ocular surface. First, we attempted to measure the tear film break-up time (as in Figure 4D). However, the tear film in the *N*-acetylcysteine-treated rabbit model was broken 1–2 s after the last blink, so that the TBUT could not be detected. Therefore, we measured the break up level to evaluate the therapeutic effect of the repetitive application of MPCP (Figure 6). The instillation of *N*-acetylcysteine decreased the lacrimal fluid volume and tear film breakup level (Figure 6). The instillation of MPCP significantly increased both the lacrimal fluid volume and tear film breakup level in comparison with the vehicle and MPC-BMA (Figure 6A,C,D). In this study, we showed that the MPCP has a strong ability to retain water on the cornea in comparison with MPC-BMA (Figure 5). The in vitro study illustrated in Figure 5 supports the in vivo study shown in Figure 6. In contrast with the results of the normal rabbits (Figure 4), the mucin levels were found to be approximately 2.5-fold greater when compared with the control (non-instillation group) (Figure 6B). Urashima et al. reported that *N*-acetylcysteine reduces the mucin-like substances in the cornea and conjunctiva and that the mucin levels are enhanced by positive feedback during the healing process [26]. Therefore, mucin production in rabbits instilled with MPCP may be overexpressed, since MPCP accelerates the normalization of the ocular surface in the dry eye model by keeping the tear film on the cornea. Taken together, we hypothesized that the MPCP would show a high affinity for lacrimal fluid on the cornea after instillation and prolonged moisture retention in the rabbit cornea, resulting in an acceleration of the improvement in tear film breakup levels. In addition, normalization of the ocular surface conditions may induce high mucin production in dry eye models.

In this study, rabbit body weight did not change with the repetitive application of MPCP over 30 d (once a day), although it remains important to assess the systemic parameters and pharmakokinetics in the rabbit model. In addition, further studies are needed to apply the MPCP as dry eye drops. In future work, we intend to measure the therapeutic effect under a combination of MPCP and commercially available eye drops, such as Mucosta^®^, Diquas^®^, Xiidra^®^ Restasis^®^, and Ikervis^®^, in a rabbit model treated with *N*-acetylcysteine.

## 5. Conclusions

In this study, we prepared a novel MPCP using MPC/stearyl methacrylate/*N*,*N*-dimethylacrylamide (50:5:45) and demonstrated that this MPCP provides high water affinity in comparison with the preexisting MPC polymer. The instillation of the MPCP prolonged the retention of lacrimal fluid in the eye and enhanced the TBUT in the rabbits. In addition, the MPCP accelerated the normalizing of the ocular surface in the dry eye model by keeping the tear film on the cornea. Moreover, the therapeutic effect of MPCP was significantly higher than that of the MPC-BMA reported previously [12]. Thus, this MPCP may provide an effective therapeutic treatment for dry eye.

## Figures and Tables

**Figure 1 pharmaceutics-13-00168-f001:**
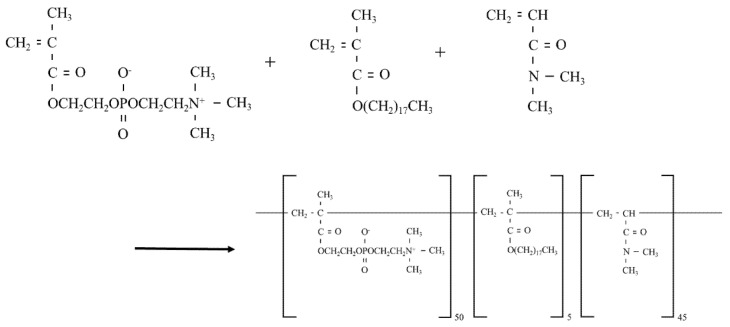
The structural formula for MPCP in this study. The MPCP based on MPC/stearyl methacrylate/*N*,*N*-dimethylacrylamide was prepared via radical polymerization. MPC—2-methacryloyloxy ethyl phosphorylcholine; MPCP—MPC polymer.

**Figure 2 pharmaceutics-13-00168-f002:**
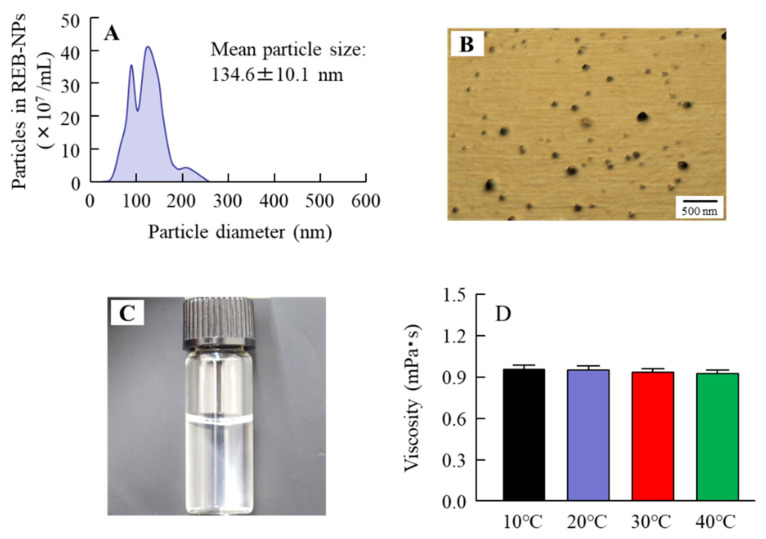
Particle size and viscosity of the MPCP. (**A**,**B**) Particle distribution (**A**) and atomic force microscopy (AFM) images (**B**) of MPCP. (**C**) Image of MPCP. (**D**) Changes in the viscosity of the MPCP at 10–40 °C. *n* = 8–10. The particle size of MPCP was 50–250 nm, and the viscosity of MPCP was 0.95 mPa·s at 20 °C. The temperature (10–40 °C) did not affect the viscosity.

**Figure 3 pharmaceutics-13-00168-f003:**
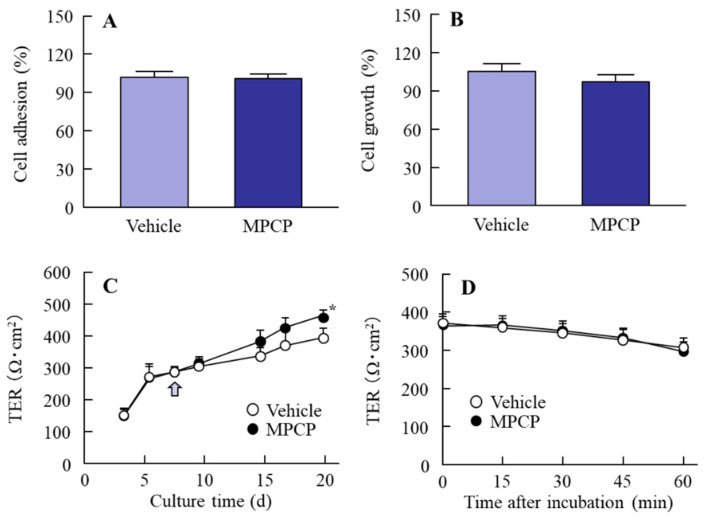
Corneal epithelial cell adhesion, growth, and stimulation of MPCP in the immortalized human corneal epithelial cell line (HCE-T cells). (**A**) Effect of MPCP on the adhesion of HCE-T cells. (**B**) Effect of MPCP on the growth of HCE-T cells. (**C**) Changes in transepithelial electrical resistance (TER) values during the formation of the HCE-T cell layer model treated with MPCP. The arrow indicates the beginning of the air–liquid interface. (**D**) Effect of MPCP on the TER values in the HCE-T cell layer model. *n* = 10–12. * *p* < 0.05 vs. the vehicle for each group. The treatment with MPCP did not affect the cell adhesion and growth of the HCE-T cells, and no stimulation was observed; however, the TER values in the HCE-T cell layer model were enhanced by the MPCP treatment.

**Figure 4 pharmaceutics-13-00168-f004:**
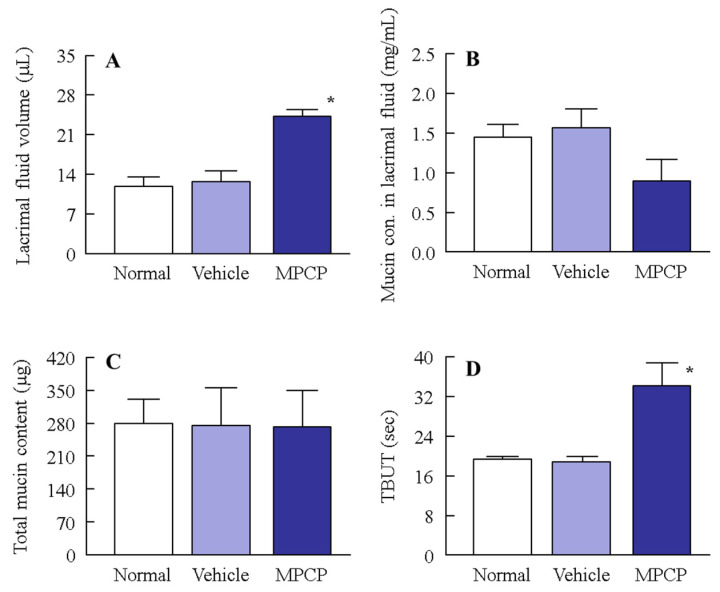
Effect of a single application of MPCP on lacrimal fluid volume, mucin levels, and tear film breakup time (TBUT) in normal rabbits. (**A**,**B**) Changes in lacrimal fluid volume (**A**) and mucin concentration in lacrimal fluid (**B**) after the application of MPCP. (**C**) Changes in total mucin levels in the lacrimal fluid after the application of MPCP. (**D**) Changes in the TBUT in rabbits instilled with the vehicle and MPCP. These measurements were performed 2 h after the instillation of MPCP (16:00). *n* = 6–8. * *p* < 0.05 vs. the vehicle for each group. Although the application of MPCP induced an increase in lacrimal fluid volume and TBUT in the rabbit eye, the total mucin contents in the rabbits instilled with MPCP were similar to those in the vehicle.

**Figure 5 pharmaceutics-13-00168-f005:**
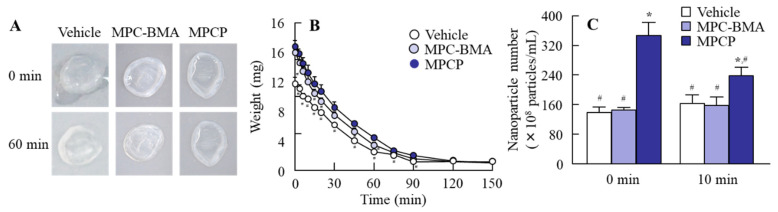
Changes in moisture retention in the cornea instilled with the vehicle, MPC-BMA, and MPCP. (**A**) Image of the cornea at 0 and 60 min after the instillation of MPCP. (**B**) Moisture retention curve in the extracted cornea instilled with the vehicle, MPC-BMA, and MPCP for 150 min. (**C**) the number of nanoparticles on the cornea at 10 min after the instillation of MPCP. Vehicle, vehicle-treated cornea. MPC-BMA, MPC-BMA-treated cornea. MPCP, MPCP-treated cornea. *n* = 5–7. * *p* < 0.05 vs. the vehicle for each group. ^#^
*p* < 0.05 vs. MPCP at 0 min. The water content in the cornea instilled with MPCP was higher than that found in the cornea instilled with the vehicle, and the time needed for water maintenance was prolonged in comparison with the vehicle and MPC-BMA. The number of MPCP particles on the cornea decreased after treatment.

**Figure 6 pharmaceutics-13-00168-f006:**
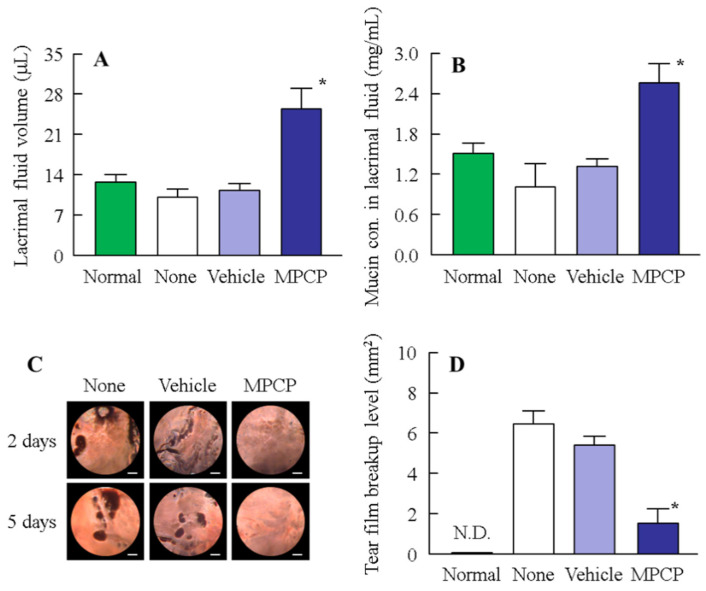
The therapeutic effect of the repetitive application of MPCP on dry eye in the *N*-acetylcysteine-treated rabbit model (dry eye model). (**A**,**B**) The effect of MPCP on the lacrimal fluid volume (**A**) and mucin levels (**B**) in the dry eye model. The mucin levels in the lacrimal fluid are expressed as the ratios of the mucin contents at the start of the experiment. (**C**) Images of the ocular surface in the dry eye model after repetitive applications of MPCP. The bar indicates 1 mm. The dark spots reflect the tear film breakup. (**D**) Effect of MPCP on tear film breakup levels in the dry eye model. Rabbits were instilled with MPCP at 14:00 h once a day for five days, and the experiments were performed at 18:00 h. This protocol was performed by following our previous reports [13]. *n* = 9–12. N.D., not detectable. * *p* < 0.05 vs. none for each group. The application of MPCP increased the volume of lacrimal fluid and normalized the decreased mucin levels in the dry eye model. The levels of tear film breakup were attenuated by the application of MPCP.

**Figure 7 pharmaceutics-13-00168-f007:**
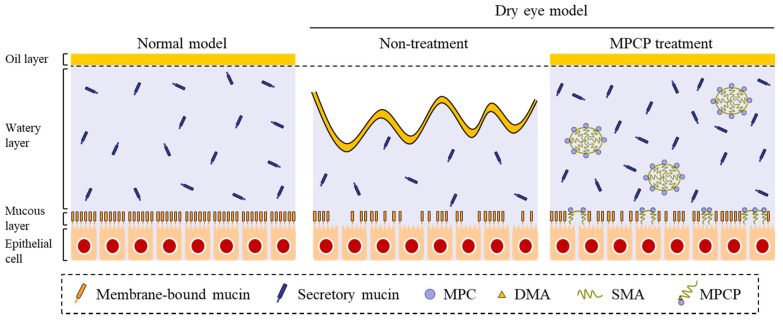
Schematic illustration for the amelioration of dry eye via the instillation of the MPCP. DMA—*N*,*N*-dimethylacrylamide; SMA—stearyl methacrylate.

## Data Availability

Not applicable.

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
