# Peer review of "MPC Polymer Promotes Recovery from Dry Eye via Stabilization of the Ocular Surface"

_pharmaceutics, 2021, doi:10.3390/pharmaceutics13020168_

Round 1
Reviewer 1 Report
The manuscript has been amended according to reviewers comments, and looks much improved. Please, check and correct the English language in the new text additions (labeled in yellow).
Author Response
We carefully revised our manuscript according to the suggestions of the reviewer 1. The details are as follows.
< Q and A for Reviewer 1>
Q1. The manuscript has been amended according to reviewers comments, and looks much improved. Please, check and correct the English language in the new text additions (labeled in yellow).
A1. Thank you very much for pointing this out. In order to respond to the reviewer’s comment, we revised these sentence.
Thank you for great comments.

Reviewer 2 Report
I have no further comments.
Author Response
We carefully revised our manuscript according to the suggestions of the reviewer 2. The details are as follows.
< Q and A for Reviewer 2>
Q1. I have no further comments.
A1. Thank you for great comments.

This manuscript is a resubmission of an earlier submission. The following is a list of the peer review reports and author responses from that submission.
Round 1
Reviewer 1 Report
The authors describe the synthesis and the use of a new synthetic polymer (PMCP) derived from a preexisting one (PMC). Hygroscopic and lubricant properties of PMC are already known, and this polymer is already present in artificial eye drops for the treatment of eye dryness. There is no adequate explanation in this paper why the authors decided to synthesize a variation of PMC, and no comparison between the properties of PMC and PMCP in terms of physico-chemical or biologic behavior. The discussion is a mere repetition of the results, and does not consider the peculiar properties of the PMCP in relation to the other different polymers used in the treatment of dry eye, thus hinting at why PMCP should be preferred to the others, at least in some occasions.
Other points of concern: there is no reference given for the corneal immortalized epithelial cell line used in this study.
There is no good description of the NAC rabbit dry eye model. In the discussion the characteristics of the model should be related to the properties of the PMCP.
Air lift culture method is used to generate a multilayer, and cells - if healthy - do not stay as monolayer. A histologic section of the filter with the cells after air lifting must be included to show their point.
The cell count method used appear to be based on formazan salts conversion, which depend on the metabolic state of the cells, and not necessarily correlates with cell number or growth. This should be considered when describing the method.
Other minor points have been indicated in the notes to the attached manuscript.

Author Response
We carefully revised our manuscript according to the suggestions of the reviewer 1, and details are as follows.
< Q and A for Reviewer 1>
Q1. There is no adequate explanation in this paper why the authors decided to synthesize a variation of PMC, and no comparison between the properties of PMC and PMCP in terms of physico-chemical or biologic behavior.
A1. The reviewer’s comment is correct. We designed MPCP structure based on 3 points. (1) The MPC, indicating the zwitterionic group, shows the hydrophilic part and gives hydrophilicity to MPCP. (2) Stearyl methacrylate including the long-chain alkyl group shows its hydrophobic part and forms hydrous polymer nano-sphare in aqueous solution. (3) Coupled with the acryl group as a highly reactive functional group, N,N-dimethylacrylamide accelerates the polymerization between MPC and stearyl methacrylate and strengthen the structure of polymer nano-sphare in aqueous solution. In addition, we measured the contact angle, velocity, moisture retention and therapeutic effect between the properties of MPC-BMA and PMCP by using a contact angle meter. In order to respond to the reviewer’s comment, we added the data and information (line 102-107, 182-188, 196-201, 243-248, 278-279, 303-309, 340-344, 358).
Q2. The discussion does not consider the peculiar properties of the PMCP in relation to the other different polymers used in the treatment of dry eye, thus hinting at why PMCP should be preferred to the others, at least in some occasions.
A2. The reviewer’s comments are very important. We compared the viscosity, contact angle and therapeutic effect between MPCP and MPC-BMA (preexisting MPC polymer). The viscosity, contact angle of MPCP was 0.92- and 0.97-fold that of MPC-BMA reported previously (preexisting MPC polymer). On the other hand, the therapeutic effect of MPCP was significantly higher than that in MPC-BMA in the dry eye rabbit model (Fig. 6). In order to respond to the reviewer’s comment, we showed that the viscosity, contact angle data of MPC-BMA, and discussed the usefulness of MPCP to compere the other different polymer (line 102-107, 196-201, 243-248, 278-279, 303-309, 340-344, 358).
Q3. There is no reference given for the corneal immortalized epithelial cell line used in this study.
A3. Thank you very much for pointing this out. The immortalized human cornea epithelial cell line (HCE-T cell) used in this study were developed by Araki-Sasaki et al. [15]. In addition, Toropainen et al. [18] developed HCE-T cell layer model based on the HCE-T cells reported by Araki-Sasaki et al. In order to respond to the reviewer’s comment, we added the reference (line 110-111, 132-133, Reference 15 and 18).
Q4. There is no good description of the NAC rabbit dry eye model. In the discussion the characteristics of the model should be related to the properties of the PMCP.
A4. Thank you for pointing out this. The N-acetylcysteine, a reducing agent and a mucolytic [22], acts on corneal and conjunctival epithelial cells and decreases mucin production and retention. Further, the instillation of 20% N-acetylcysteine caused a decrease in the mucin layer in the conjunctiva and cornea, desquamation of conjunctival and corneal epithelial cells, elimination of microvilli [23] and the thickness of the tear fluid layer [24]. Because of these actions, the rabbit model treated with N-acetylcysteine is widely applied as a mucin-reduced model. On the other hand, the water content in cornea instilled with MPCP was higher than that found in cornea instilled with the vehicle, and the time for water maintenance was prolonged in comparison with the vehicle and MPC-BMA (Fig. 5). These results suggested that the MPCP may provide the alternative role for reduced mucin, and normalize the ocular surface environment. Taken together, we used the N-acetylcysteine rabbit model to evaluate the therapeutic effect of MPCP. In order to respond to the reviewer’s comment, we added the MPC-BMA data, and showed these contents in the Discussion (line 247-248, 339-344, Figure 5).
Q5. Air lift culture method is used to generate a multilayer, and cells - if healthy - do not stay as monolayer. A histologic section of the filter with the cells after air lifting must be included to show their point.
A5. The reviewer’s comment is correct. The immortalized human cornea epithelial cell line (HCE-T cell) used in this study were developed by Araki-Sasaki et al. [15]. In addition, Toropainen et al. [18] developed HCE-T cell layer model based on the HCE-T cells reported by Araki-Sasaki et al. The histologic section was reported previous reports [18], and the barrier properties of HCE-T cell layer model are very similar to those of excised rabbit corneas, making HCE-T cell layer model an alternative corneal substitute for ocular drug delivery studies. In this study, we used the HCE-T cell layer model developed by Araki-Sasaki et al. and Toropainen et al. [15, 18], and the transepithelial electrical resistance (TER) was similar to the data of Toropainen et al. [18]. Therefore, we think that the HCE-T cell layer model is appropriate. On the other hand, in this study, the “monolayer” was used to mean that it consists of one cell (only HCE-T cells), and the word may be misleading to the reader. In order to respond to the reviewer’s comment, we added the information and reference, and corrected to “cell layer model” from “monolayer”. Thank you very much for pointing this out. (line 110-111, 132-133, Reference 15 and 18).
Q6. The cell count method used appear to be based on formazan salts conversion, which depend on the metabolic state of the cells, and not necessarily correlates with cell number or growth. This should be considered when describing the method.
A6. The reviewer’s comments are very important. We also measured the cell number by counting under the microscope, and the cell number was not difference between of vehicle and MPCP. We think that the combination of data for cell number and Cell Count Reagent SF reflect the cell adhesion and growth. In order to respond to the reviewer’s comment, we added the data for cell number in the Result (line 117-121, 128-129).
Q7. Other minor points have been indicated in the notes to the attached manuscript.
A7. The reviewer’s comment is correct. In order to respond to the reviewer’s comment, we completely corrected and added reference. Thank you very much for pointing this out.
Thank you for great comments.

Reviewer 2 Report
This is an Interesting application of the MCP material in the treatment of dry eye. Good evaluation of their technique and experiments are sound. There are a couple of issues in their approach.
1- Why are the authors using N-acetylcysteine to make a dry eye in Rabbit? As far we know, this isn't a go to reagent to cause dry eye. Research groups use it as an antioxidant and in some cases of dry eye in humans, it is used as treatment. Unless, they used it because of the chemistry of the structure does not interrupt with the function of the MCP. I'm not sure, but still using it isn't an established method of causing dry eye. 2- Why did the authors look at Mucin? I understand the function of the mucin but there are more important proteins to look at in terms of assessing dry eye and treatment effect. Mucin is mostly concerned with goblet cells and conjunctival secretions. Maybe the authors should look at secretions of the lacrimal glands and meibomian glands to assess recovery of dry eye by treatments.Author Response
We carefully revised our manuscript according to the suggestions of the reviewer 2, and details are as follows.
< Q and A for Reviewer 2>
Q1. Why are the authors using N-acetylcysteine to make a dry eye in Rabbit? As far we know, this isn't a go to reagent to cause dry eye. Research groups use it as an antioxidant and in some cases of dry eye in humans, it is used as treatment.
A1. Thank you very much for pointing this out. Sheffner et al. reported that the N-acetylcysteine, a reducing agent and a mucolytic, acts on corneal and conjunctival epithelial cells and decreases mucin production and retention [22]. Further, it was known that the instillation of 20% N-acetylcysteine caused a decrease in the mucin layer in the conjunctiva and cornea, desquamation of conjunctival and corneal epithelial cells, elimination of microvilli [23] and the thickness of the tear fluid layer [24]. Thus, repetitive instillation caused the dry eye with the decrease in mucin level. Following these previous reports, the dry eye model using rabbit was conducted by the instillation (30 µL) of 10% N-acetylcysteine at 6 times per day (9:00, 11:00, 13:00, 15:00, 17:00 and 19:00) in this study. In order to respond to the reviewer’s comment, we added the detail protocol (how many administrations were necessary to induce dry eye) in the Materials and Methods and Discussion (line 80-81, 348-356).
Q2. Why did the authors look at Mucin? I understand the function of the mucin but there are more important proteins to look at in terms of assessing dry eye and treatment effect. Mucin is mostly concerned with goblet cells and conjunctival secretions. Maybe the authors should look at secretions of the lacrimal glands and meibomian glands to assess recovery of dry eye by treatments.
A2. The reviewer’s comments are very important. In the in vitro study (Fig. 5), we found that the water content in cornea instilled with MPCP was higher than that found in cornea instilled with the vehicle, and the time for water maintenance was prolonged in comparison with the vehicle and MPC-BMA. These actions of MPCP are similar to the role of mucin in the ocular surface. Therefore, we evaluated the therapeutic effect of MPCP on the dry eye by using the N-acetylcysteine-dry eye model with the low mucin level, and measured the changes in mucin levels. From these findings, we think that the measurement of mucin levels in the N-acetylcysteine-dry eye model is important to demonstrate the therapeutic effect of MPCP. Thank you very much for pointing this out.
Thank you for great comments.

Reviewer 3 Report
This is an interesting manuscript reporting on experimental studies with a MPC polymer which might in the future be used in dry eye therapy. Please find my questions and comments below:
- In my opinion, it would be of interest to state more details about MPC in the introduction and for which other purposes it is also used, and whether these other uses are only under study or if there are already approved applications.
- Did the authors also assess systemic parameters or pharmakokinetics in the rabbit model?
Author Response
We carefully revised our manuscript according to the suggestions of the reviewer 3, and details are as follows.
< Q and A for Reviewer 3>
Q1. In my opinion, it would be of interest to state more details about MPC in the introduction and for which other purposes it is also used, and whether these other uses are only under study or if there are already approved applications.
A1. The reviewer’s comments are very important. So far, soft contact lens containing the MPC was approved by Food and Drug Administration (FDA) and Pharmaceuticals and Medical Devices Agency (PMDA). Hall et al. reported that the dehydration of this soft contact lens showed significantly less than other conventional lenses [11]. On the other hand, the eye drop, soft contact lens care products, oral care product and cosmetics containing MPC-BMA polymer is approved by PMDA, too. In order to respond to the reviewer’s comment, we added the information in the Introduction (line 55-59, Reference 11).
Q2. Did the authors also assess systemic parameters or pharmakokinetics in the rabbit model?
A2. Thank you for pointing out this. The body weight was not changed in the rabbit with repetitive application of MPCP for 30 d. From this reason, we think that the repetitive application of MPCP is not provide the highly systemically toxicity. On the other hand, we don’t have the data for pharmacokinetics of MPCP in the rabbit model. Therefore, we add that the necessary to measure the pharmakokinetics in the rabbit model in the further study. In order to respond to the reviewer’s comment, we added these contents in the Results and Discussion (line 367-369).
Thank you for great comments.

Round 2
Reviewer 1 Report
Authors have tried to answer to the points raised in the previous review, and have added some more data in the revised presentation of the manuscript.
However, the impression still remains that the new molecule MPCP has some merit towards treatment of dryness, but it is difficult to judge it in comparison to existing treatments.
New data are reported about the wettability measured as the contact angle, showing some superiority of MPCP over MPC-BMA.
Figure 3 (unchanged) still shows a significant increase of TEER only after 20 days of air-lift culture, with no reference to what would happen with MPC-BMA.
Same for figure 4, showing a significant increase in normal rabbits only for BUT, and no idea what happens with MPC-BMA. Also, normal control values should be reported in each panel.
Figure 5 is the only including a reference behavior with MPC-BMA, however in a way that is hardly intelligible, since the statistics in panel B is not readable: it is important to show the difference between vehicle and treatments, and also between the two treatments, to show whether they exert similar effects, or one is better than the other.
Figure 6 shows the results on the dry eye rabbit model system. It looks that there are no significant differences between NAC-treated and -untreated rabbits (so there is no evident dry eye), and MPCP improves all the parameters, like in normal rabbits. Parameters should be shown all as real values, and not percent (this can be given in the results section in writing), and normal values should be present in all panels.
The discussion is again mostly a repetition of the results. This is because no real comparisons have been done between MPCP and MPC-BMA, that allow some speculation on the different molecular mechanisms involved with the two molecules. The only interesting point was raised in lines 358-359, and it deserves to be discussed more thoroughly.
Author Response
We carefully revised our manuscript according to the suggestions of the reviewer 1, and the manuscript has been checked and edited by MDPI English editing service again. The details are as follows.
< Q and A for Reviewer 1>
Q1. Figure 3 (unchanged) still shows a significant increase of TEER only after 20 days of air-lift culture, with no reference to what would happen with MPC-BMA.
A1. Thank you for pointing out this. We previously reported that MPC-BMA dose not affect the cell adhesion and growth of the HCE-T cells, with little toxicity in HCE-T cells [19]. In order to respond to the reviewer’s comment, we added these contents and reference (line 322-324, Reference 19).
Q2. Same for figure 4, showing a significant increase in normal rabbits only for BUT, and no idea what happens with MPC-BMA. Also, normal control values should be reported in each panel.
A2. Thank you very much for pointing this out. The lacrimal fluid volume (18.4±1.9 µL, N=5) and TBUT (24.1±2.0 sec, N=5) in the rabbits instilled with MPC-BMA tended to increase, although both the lacrimal fluid volume and TBUT were significantly lower than those in the MPCP. In order to respond to the reviewer’s comment, we added these data, and added the normal control values in each panel (line 247-250, Figure 4).
Q3. Figure 5 is the only including a reference behavior with MPC-BMA, however in a way that is hardly intelligible, since the statistics in panel B is not readable: it is important to show the difference between vehicle and treatments, and also between the two treatments, to show whether they exert similar effects, or one is better than the other.
A3. The reviewer’s comment is correct. In order to respond to the reviewer’s comment, we mentioned that the MPC-BMA also prolonged the time needed for water maintenance in comparison with the vehicle, but, this ability was significantly lower than that in MPCP (line 254-257, 355-359).
Q4. Figure 6 shows the results on the dry eye rabbit model system. It looks that there are no significant differences between NAC-treated and -untreated rabbits (so there is no evident dry eye), and MPCP improves all the parameters, like in normal rabbits. Parameters should be shown all as real values, and not percent (this can be given in the results section in writing), and normal values should be present in all panels.
A4. The reviewer’s comments are very important. We showed the parameters as real values, and added the normal values in all panels. Due to these change, we showed that the tear film breakup was not detected in the normal rabbit, although, the tear film breakup in NAC-treated rabbits was detected 5 d after NAC treatment. Therefore, we think that the NAC-treated rabbit is dry eye model. In order to respond to the reviewer’s comment, we added the data and contents (Figure 6).
Q5. The discussion is again mostly a repetition of the results. This is because no real comparisons have been done between MPCP and MPC-BMA, that allow some speculation on the different molecular mechanisms involved with the two molecules. The only interesting point was raised in lines 358-359, and it deserves to be discussed more thoroughly.
A5. Thank you very much for pointing this out. In order to respond to the reviewer’s comment, we added the data and reference [19] of MPC-BMA, and mentioned the comparisons between MPCP and MPC-BMA (the therapeutic effect of MPCP was higher than that in MPC-BMA). In the Fig. 5, we showed that the MPCP has a strong ability to retain water on the cornea in comparison with MPC-BMA. The in vitro study of illustrated in Fig. 5 supports the in vivo study shown in Fig. 6. We mentioned these contents, and we think that these revise enhanced the quality in this manuscript. Thank you for pointing out this (line 322-324, 355-363, 378-380, Reference 19).
Thank you for great comments.

Reviewer 2 Report
Accept
Author Response
Thank you for great comment.
Round 3
Reviewer 1 Report
In this second revision the quality of the manuscript is improved. With a better quality of results presentation and description some more criticisms have emerged:
Paragraph 2.11: Usually, stability of the tear film is measured by the tear film break up time (as in figure 4D). Authors chose to use also another parameter based on the area of dried spots, that they call break up level. A better description of this parameter must be introduced: at which time (how long after the last blink) it is calculated, and how.
Figure 4: Insert statistic values in fig. 4 A and B (how it is, it looks that MPCP does not induce significative variations). Explain how lacrimal fluid volume from Schirmer test, and total mucin content are calculated.
Figure 6A: The dry eye model was induced by 10% NAC, and both Schirmer test and mucin concentration showed no decrease with respect to normal controls. Only difference is with break up level, which however is not defined in terms of timing: each picture must report the timing after blinking when the photo was taken.
Figure 6B: Mucin level should be mucin concentration (mg/mL). In this experiment mucin concentration appears much higher also with respect to normal controls, which is opposite to what shown in Figure 4B. How could this be?
Discussion, lines 350-353: Authors say: against all expectations... I would say that if lacrimation is increased and total mucin content remains stable (Figure 4 A and C), then mucin concentration is expected to decrease!
In general, the discussion should be more straightforward, be reduced in size and point to and explain the and justify the advantages of the new synthetic polymer.